

# Potential COVID-19 papain-like protease PL^{pro} inhibitors: repurposing FDA-approved drugs

Valentina L. Kouznetsova[1], Aidan Zhang[2], Mahidhar Tatineni[1], Mark A. Miller[1] and Igor F. Tsigelny[1,3,4]

[1] San Diego Supercomputer Center, University of California, San Diego, La Jolla, CA, USA
[2] REHS Program at San Diego Dupercomputer Center, University of California, San Diego, La Jolla, CA, USA
[3] Department of Neurosciences, University of California, San Diego, La Jolla, CA, USA
[4] Science, CureMatch Inc., San Diego, CA, USA

## ABSTRACT

Using the crystal structure of SARS-CoV-2 papain-like protease (PL^{pro}) as a template, we developed a pharmacophore model of functional centers of the PL^{pro} inhibitor-binding pocket. With this model, we conducted data mining of the conformational database of FDA-approved drugs. This search identified 147 compounds that can be potential inhibitors of SARS-CoV-2 PL^{pro}.
The conformations of these compounds underwent 3D fingerprint similarity clusterization, followed by docking of possible conformers to the binding pocket of PL^{pro}. Docking of random compounds to the binding pocket of protease was also done for comparison. Free energies of the docking interaction for the selected compounds were lower than for random compounds. The drug list obtained includes inhibitors of HIV, hepatitis C, and cytomegalovirus (CMV), as well as a set of drugs that have demonstrated some activity in MERS, SARS-CoV, and SARS-CoV-2 therapy. We recommend testing of the selected compounds for treatment of COVID-19

## INTRODUCTION

Coronaviruses have caused the outbreak of several deadly respiratory diseases since the turn of the 21st century, such as the severe acute respiratory syndrome (SARS) in 2002 and the Middle East respiratory syndrome (MERS) in 2012, in addition to the recent COVID-19 pandemic, which has claimed more than 489,000 lives with over 9.5 million confirmed cases worldwide. Despite the profound impact of these viral outbreaks on public health and the economy, effective vaccines have not been found for either SARS or MERS viruses. In view of the ongoing pandemic, and the absence of vaccines, there is an immediate need to find drugs to treat patients.

Viral proteases are an attractive target for drug development. Viral proteases are essential for replication, and are unique to each virus, thus offering the potential for highly

Corresponding author
Igor F. Tsigelny, itsigeln@ucsd.edu

specific treatments that produce minimal toxic side effects. Viral protease inhibitors such as indinavir targeting a single protease in HIV-1, ritonavir targeting the single proteases in HIV-1 and HIV-2, and boceprevir targeting NS3 protease of HCV have been used to effectively treat a variety of viral infections (*Anderson et al., 2009*). For coronaviruses, host protease TMPRSS2 provide a possible target where protease inhibitors can prevent viral entry (*Hoffmann et al., 2020*; *Simmons et al., 2005*; *Zhou et al., 2015*). On the other hand, two viral proteases, papain-like protease (PL^pro) and chymotrypsin-like protease (3CL^pro, aka main protease) are also attractive as druggable targets (*Vuong et al., 2020*; *Ma et al., 2020*). Both proteases are highly conserved specific nsps: nsp5 for 3CL^pro and nsp3 for PL^pro. Nsp3 is a large (200,000 kDa) multi-domain polypeptide that provides the membrane anchored scaffolding structure required for the replication/transcription complex of coronaviruses (*Lei, Kusov & Hilgenfeld, 2018*). In addition to PL^pro, the C-terminus of nsp3 contains transmembrane domains that anchor the protein and a dsDNA, unwinding/RNA binding domain that is essential for replicase activity (*Neuman, 2016*). Residing within the 213-kDa, membrane-associated replicase product nsp3, the SARS-CoV PL^pro is responsible for cleaving junctions spanning nsp1 to nsp4 (*Devaraj et al., 2007*). Studies of Harcourt and colleagues (*Harcourt et al., 2004*) revealed that PL^pro can cleave at the three predicted cleavage sites and that it requires membrane association to process the nsp3/4 cleavage site.

It is a particularly attractive drug target because it plays an essential role in processing the viral polyproteins to create the mature nsp3, as well as helping the coronavirus evade host immune response via competitive interaction with ubiquitin and ISG15 on host-cell proteins (*Lei, Kusov & Hilgenfeld, 2018*; *Wu et al., 2020*; *Báez-Santos, St. John & Mesecar, 2015*; *Lei et al., 2014*). Although no protease inhibitors are currently available for treatment of SARS, MERS, or COVID-19, studies of inhibitors of the MERS, SARS-CoV, and SARS-CoV-2 PL^pro are underway and reports have appeared that such protease inhibitors can prevent SARS-CoV replication in cultured cells (*Dyall et al., 2014*; *Báez-Santos, St. John & Mesecar, 2015*; *Ratia et al., 2008*; *Lee et al., 2015*; *Akaji et al., 2011*).

In view of the urgent need for effective treatments and the high cost of developing new drugs (both in terms of time and resources), repurposing FDA-approved drugs is an efficient strategy for identifying drug candidates that can be used immediately in the COVID-19 pandemic (*Tan et al., 2004*; *Kouznetsova, Huang & Tsigelny, 2020*). In a previous report, we (*Kouznetsova, Huang & Tsigelny, 2020*) and others (*Kandeel & Al-Nazawi, 2020*; *Arya et al., 2020*; *Liu & Wang, 2020*; *Plewczynski et al., 2007*; *Ton et al., 2020*) have used molecular modeling studies to identify FDA-approved drugs and other compounds (*Arya et al., 2020*; *Liu & Wang, 2020*; *Ton et al., 2020*; *Alamri, Tahir ul Qamar & Alqahtani, 2020*) that are predicted to bind to 3CL^pro. The list of potential inhibitors includes bleomycin, mithramycin, and goserelin, as well as others that may be effective (*Kouznetsova, Huang & Tsigelny, 2020*). Here we report a similar screen of FDA-approved drugs for potential inhibitors of SARS-COV-2 PL^pro using the recently reported structure of SARS-CoV-2 PL^pro (PDB ID: 6W9C) (*Zhang et al., 2020*; *Osipiuk et al., 2020*).

## METHODS

### Pharmacophore design and use

Analyzing a pocket, we elucidated a majority of possible interactions between PL$^{pro}$ (PDB ID: 6W9C) and a potential ligand for developing a protein-based pharmacophore model with potential fictional centers that would bind to the residues in the pocket (Fig. 1A). Resolution of the protein structure used in the study is 2.7 Å. From our experience such a resolution is not the best but sufficient for the pharmacophore-based modeling. Using Molecular Operating Environment (MOE; CCG, Montreal, QC, Canada), we constructed two pharmacophore models including 10 features (Pha01) and 10 features with excluded volume $R$ = 1.3 Å (Pha02): two donors, two donors or acceptors, one hydrophobic, and five hydrophobic or aromatic features (Fig. 1A). Based on developed pharmacophores to select potential drug-candidates, we conducted a pharmacophore search with both pharmacophore models on our conformational database (DB) of FDA-approved drugs, containing around 2,500 drugs and 600,000 conformations. Searches were provided using pharmacophores partial match: 8 of 10 features for Pha01 and 7 of 10 features for Pha02. Search results of Pha01 (Search 1) identified 405 compounds with 63,821 conformations while Pha02 (Search 2) identified 857 compounds with 224,609 conformations. We selected 84 and 77 compounds from Search 1 and 2 respectively based on a number of H-bonds and hydrophobic interactions in the best docking pose. Because some compounds appeared in both searches, we eliminated duplicate compounds, resulting in a total of 147 unique drugs. Then we clustered the selected 147 compounds, using MOE Database Viewer with a fingerprint GpiDAPH3 and similarity–overlap parameter SO = 42% to elucidate the common structure–functional features of the groups of compound to enhance further drug development.

### Docking of drug conformers using the supercomputer Comet

For docking the selected compounds, we used the crystal structure of the SARS-CoV-2 PLP (PDB ID: 6W9C). A binding pocket was defined based on the known residues of the S3/S4 binding pocket site of SARS-CoV-2 PLP. Docking of the selected compounds was done using Autodock Vina. Conformers of each of the selected compounds were generated using OpenBabel. However, since Autodock Vina does not support docking compounds that include boron atoms (i.e., bortezomib), each boron atom in the conformers of bortezomib was replaced with carbon atoms due to their similar size. The random control compounds were selected by a 79-compound, simple-random subset of all the ZINC DB compounds; these were docked with PL$^{pro}$ in the same processes. Likewise, the conformers of the compound with ID: ZINC001779539170 had their silicon atom replaced with carbon due to Autodock Vina's restraints regarding supported atoms.

The Comet supercomputer at the San Diego Supercomputer Center (SDSC) was primarily used for two parts of the analyses: (1) conversion of files in the pdb format to the pdbqt format, using the Open Babel software (version 2.4.1), and (2) all the docking computations using the AutoDock Vina software (version 1.1.2). We outline the system configuration and the analyses workflow details below.

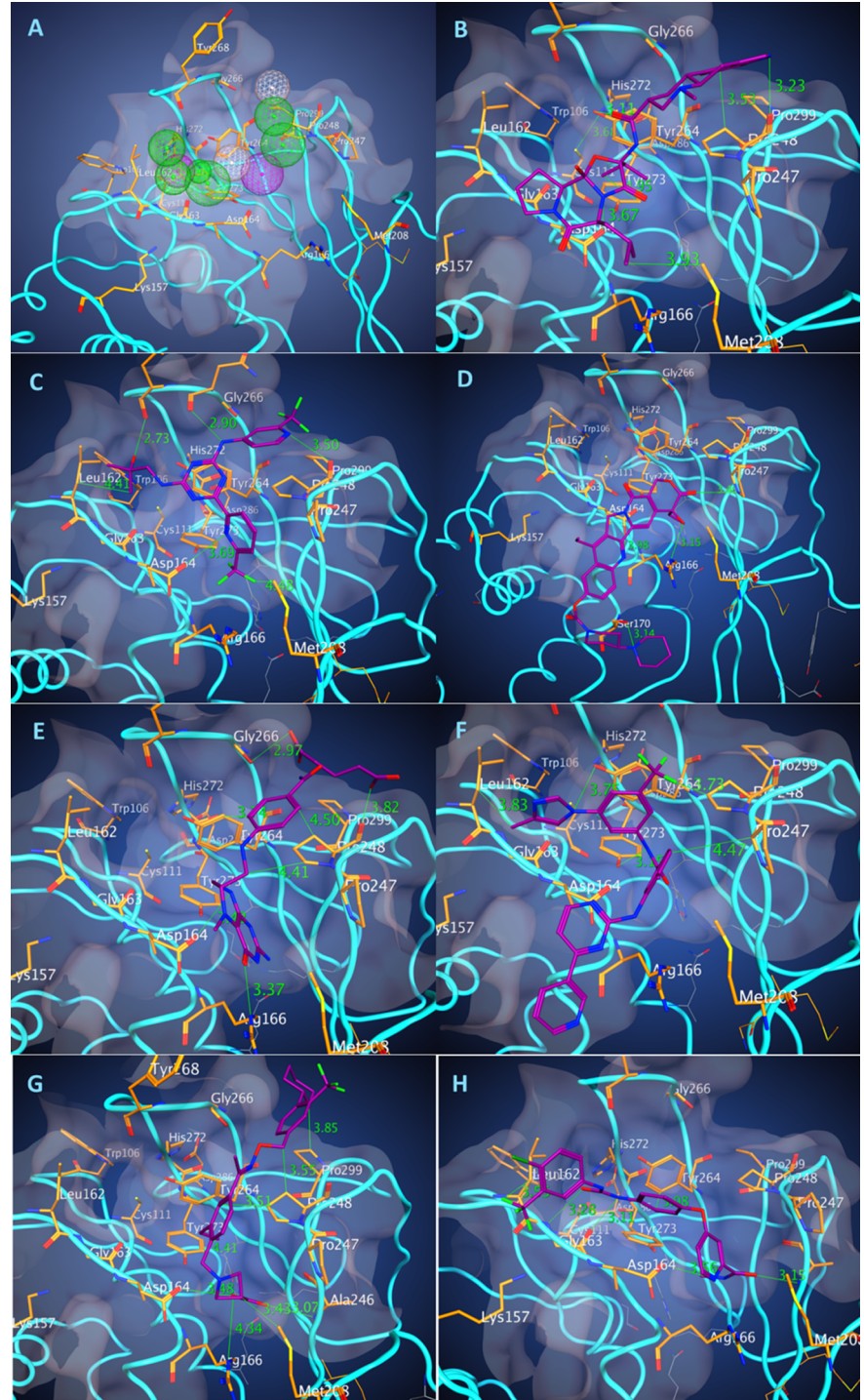

**Figure 1 Pharmacophore of PLpro binding pocket and the binding poses of the best energy docked molecules.** The model of pharmacophore (A) contains 10 functional centers: two donors, two donor/acceptor centers, one hydrophobic center, and five hydrophobic and aromatic centers. Binding poses of the drugs with the best scores. (B) Dihydroergocryptine, docking free energy (DFE) = −8.0 kcal/mol. (C) Enasidenib, (DFE) = −8.1 kcal/mol. (D) Irinotecan, (DFE) = −8.5 kcal/mol. (E) Levomefolic acid, (DFE) = −8.4 kcal/mol. (F) Nilotinib, (DFE) = −9.3 kcal/mol. (G) Siponimod, (DFE) = −8.0 kcal/mol. (H) Sorafenib, (DFE) = −8.0 kcal/mol.                               

### The Comet supercomputing system

Comet is an NSF funded cluster (NSF grant: ACI #1341698) designed by Dell and SDSC delivering 2.76 peak petaflops. It features Intel Haswell processors with AVX2, Mellanox FDR InfiniBand interconnects, and Aeon storage (*Moore et al., 2014*). There are 1,944 standard compute nodes and 72 GPU nodes. The standard compute nodes consist of Intel Xeon E5-2680v3 (Haswell) processors, 128 GB DDR4 DRAM (64 GB per socket), and 320 GB of SSD local scratch memory. The GPU nodes contain four NVIDIA GPUs each. There are four large memory nodes containing 1.5 TB of DRAM and four Haswell processors each. All the computations for this paper were conducted on the standard compute nodes and made extensive use of the local scratch filesystems.

### File conversion and docking workflow

The first step in the computational workflow on Comet was to convert 385,193 pdb files of drug conformers into the pdbqt format. The files were contained in 27 zip files and the jobs were simultaneously run on Comet (one zip file in each job). The zip files were extracted to the local SSD based file system to reduce IO loads, converted to pdbqt files in the same location, and then the results were archived in a zip file. With the local SSD approach, all the conversion jobs were completed in less than 20 min.

The AutoDock Vina software was used to dock a total of 490,678 drug conformers using computations on Comet. The local SSD approach was used again to mitigate IO loads on the main filesystem. The docking tasks were split up into separate jobs (that were run simultaneously) with 3,000–4,000 drug conformers docked in each job. All the individual docking computations were conducted using eight cores (The parallelism is limited by the exhaustiveness parameter, set to 8 for the analysis) and scaling tests showed an excellent parallel efficiency of 93.2% (*World Health Organization, 2020*).

## RESULTS

Among the compounds selected by the pharmacophore search of FDA-approved drug DB, we identified two clusters (A and B) containing 20 compounds; 3 clusters (C, D, and E) containing 9, 5, and 10 compounds correspondingly; 2 clusters (F and G) with 4, and 3 clusters (H, I, and J) with 3 compounds; along with 10 two-compounds clusters and 46 not clustered single compounds. Compounds in clusters A–G are listed in Table 1, other compounds can be found in Supplemental Materials (Table S1). Flexible alignment of clusters B and C were used to illustrate compounds' common features (Fig. 2).

Interesting to note that this selection contained the best docking energy drug nilotinib that showed activity against SARS-CoV.

To define the putative best binding drugs, we conducted docking of multiple conformers of drugs selected from a pharmacophore-based search and of random compounds to the binding site of COVID-19 PL$^{pro}$. The random control compounds were selected by a 79-compound, simple-random subset of the ZINC DB of drug-like

**Table 1 Drug-candidates clustered by fingerprint similarity–overlap alignment.**

**Cluster**

| A | B | C | D | E | F | G |
|---|---|---|---|---|---|---|
| Alclometasone | Abemaciclib | Bilastine | Dipyridamole | Acebutolol | Isoetharine | Lactulose |
| alpha-Tocopherol acetate | Bosentan | Darifenacin | Enoxacin | Atenolol | Isoxsuprine | Micronomicin |
| Bimatoprost | Cefdinir | Droperidol | Gatifloxacin | Betaxolol | Nylidrin | Netilmicin |
| Boceprevir | Cefmenoxime | Fluspirilene | Gemifloxacin | Bisoprolol | Protokylol | Tobramycin |
| Buprenorphine | Cefmetazole | Haloperidol | Moxifloxacin | Celiprolol | | |
| Calcitriol | Cefotaxime | Iloperidone | | Esmolol | | |
| Diflorasone | Cefotiam | Loperamide | | Metipranolol | | |
| Dihydroergocryptine | Cephaloglycin | Ropinirole | | Metoprolol | | |
| Flunisolide | Copanlisib | Ziprasidone | | Nadolol | | |
| Fluocinolone acetonide | Dasatinib | | | Propafenone | | |
| Ibutilide | Dicloxacillin | | | | | |
| Iloprost | Doxazosin | | | | | |
| Lapyrium | Enasidenib | | | | | |
| Lovastatin | Flucloxacillin | | | | | |
| Methyl undecenoyl leucinate | Gefitinib | | | | | |
| Retapamulin | Latamoxef | | | | | |
| Ritonavir | Nilotinib | | | | | |
| Travoprost | Prazosin | | | | | |
| Vitamin E Succinate | Riociguat | | | | | |
| Zucapsaicin | Vemurafenib | | | | | |

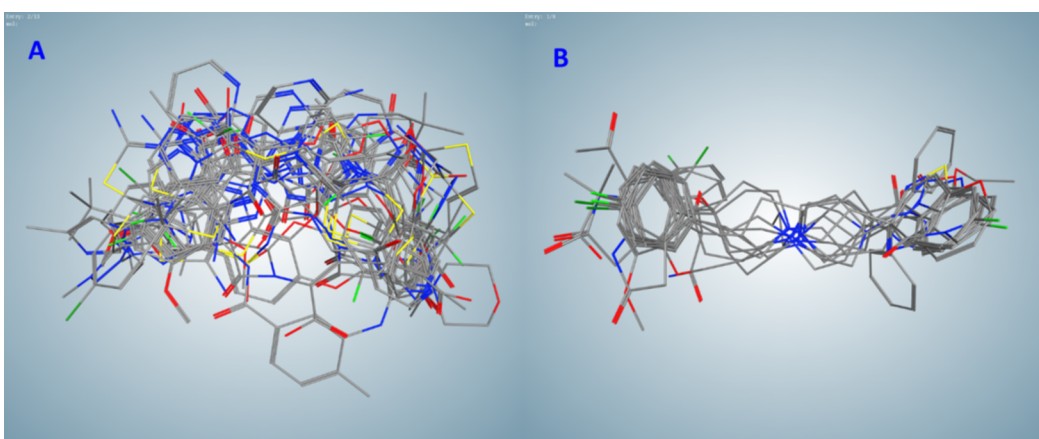

**Figure 2 Flexible alignments of compounds in clusters selected by the pharmacophore-based search of possible drug-candidates in the conformational database of FDA-approved drugs having the best docking energies.** (A) Cluster B (20 compounds). (B) Cluster C (9 compounds).

compounds. For docking the selected compounds, we used the same crystal structure of the SARS-CoV-2 (Protein Data Bank entry, 6W9C) imported into MOE. A S3/S4 pocket site was defined, which included the following residues: K157, L162, G163, D164, R166,

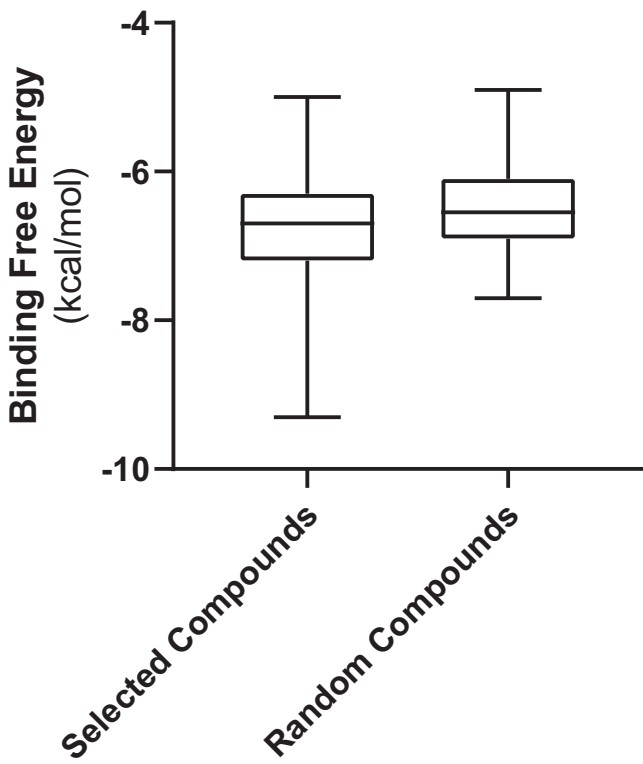

**Figure 3 Free energies of docking interactions of selected and random compounds with PL^pro.**
Minimal energies of the selected and random compounds are −9.3 and −7.7 kcal/mol respectively.

P247, P248, Y264, G266, Y268, and P299. Conformers of each of the selected compounds were generated with OpenBabel before being docked with AutoDock Vina.

Figure 3 shows the values of docking free energies of the selected and random compounds. The energies of interaction with PL^pro are shown in Table 2. One can see that drugs of clusters 2 and 5 are at the top of the table. Note that the binding pocket of PL^pro is not very specific and contains a number of hydrophobic binding centers; that is why binding energies are not overwhelmingly better than those of random compounds (Fig. 3). At the same time, we want also note that the values of energies in the table can be used with discretion. Binding positions of ligands in the pockets of proteins in many cases do not have minimal energies.

## DISCUSSION

Based on the crystal structure of SARS-CoV-2 PL^pro (PDB ID: 6W9C), we developed two pharmacophore models of the binding pocket of this protein. Using these models, we browsed our conformational database of FDA-approved drugs and obtained 147 hits that were clusterized for selecting the most promising candidates and then used for multi-conformational docking to the PL^pro pocket. The drug list obtained includes inhibitors of HIV, hepatitis C, and cytomegalovirus (CMV), as well as a set of drugs that demonstrated some activity in MERS, SARS-CoV, and SARS-CoV-2 therapy.

**Table 2 List of docked compounds sorted by their energies of interaction with COVID-19 papain-like protease in the docked positions.**

| Drug name | DFE* energy | Cluster | Drug name | DFE* energy | Cluster |
|---|---|---|---|---|---|
| Nilotinib | −9.3 | B | Losartan | −7.3 | aa |
| Irinotecan | −8.5 | S | Tolvaptan | −7.3 | S |
| Levomefolic acid | −8.4 | S | Darifenacin | −7.3 | C |
| Enasidenib | −8.1 | B | Flunisolide | −7.3 | A |
| Siponimod | −8.0 | S | Alvimopan | −7.2 | hh |
| Sorafenib | −8.0 | S | Iloperidone | −7.2 | C |
| Dihydroergocryptine | −8.0 | A | Indacaterol | −7.2 | S |
| Abemaciclib | −7.9 | B | Mirabegron | −7.2 | S |
| Ziprasidone | −7.9 | C | Ximelagatran | −7.2 | S |
| Pemetrexed | −7.8 | hh | Droperidol | −7.2 | C |
| Doxazosin | −7.8 | B | Ertapenem | −7.2 | jj |
| Axitinib | −7.7 | S | Ivacaftor | −7.1 | S |
| Indinavir | −7.7 | S | Loperamide | −7.1 | C |
| Lymecycline | −7.7 | S | Flibanserin | −7.1 | S |
| Methysergide | −7.7 | I | Brexpiprazole | −7.0 | C |
| Rutin | −7.7 | S | Cefmenoxime | −7.0 | B |
| Vemurafenib | −7.7 | B | Latamoxef | −7.0 | B |
| Glyburide | −7.7 | dd | Olmesartan | −7.0 | aa |
| Trabectedin | −7.6 | S | Bilastine | −6.9 | C |
| Dasatinib | −7.6 | B | Bosentan | −6.9 | C |
| Methylergonovine | −7.5 | I | Cefdinir | −6.9 | C |
| Riociguat | −7.5 | B | Cefotaxime | −6.9 | B |
| Fluocinolone | −7.5 | A | Prazosin | −6.9 | B |
| Fluspirilene | −7.5 | C | Retapamulin | −6.9 | A |
| Isavuconazole | −7.4 | S | Ritonavir | −6.9 | A |
| Manidipine | −7.4 | ii | Sulfasalazine | −6.9 | S |
| Regadenoson | −7.4 | S | Topotecan | −6.9 | H |
| Glimepiride | −7.4 | dd | Copanlisib | −6.9 | B |
| Canagliflozin | −7.3 | bb | Diflorasone | −6.9 | A |
| | | | Gemifloxacin | −6.9 | H |

**Notes:**
* Docking free energy.
S, single compound cluster.

We developed a pharmacophore model of the binding pocket site S3/S4 of COVID-19 PL$^{pro}$ then conducted multi-conformational docking of these drug compounds to this site for ranging the potential inhibitors selected by pharmacophore-based search. We also conducted clusterization of the selected compounds based on their pharmacophores 3D profiles to elucidate the common features for further drug design, and compared the docking results for the selected drug compounds with the docking results of random compounds to evaluate the area of significance in the values of binding energies. We note that the pharmacophore-based selection is a very powerful tool so even the drugs with the

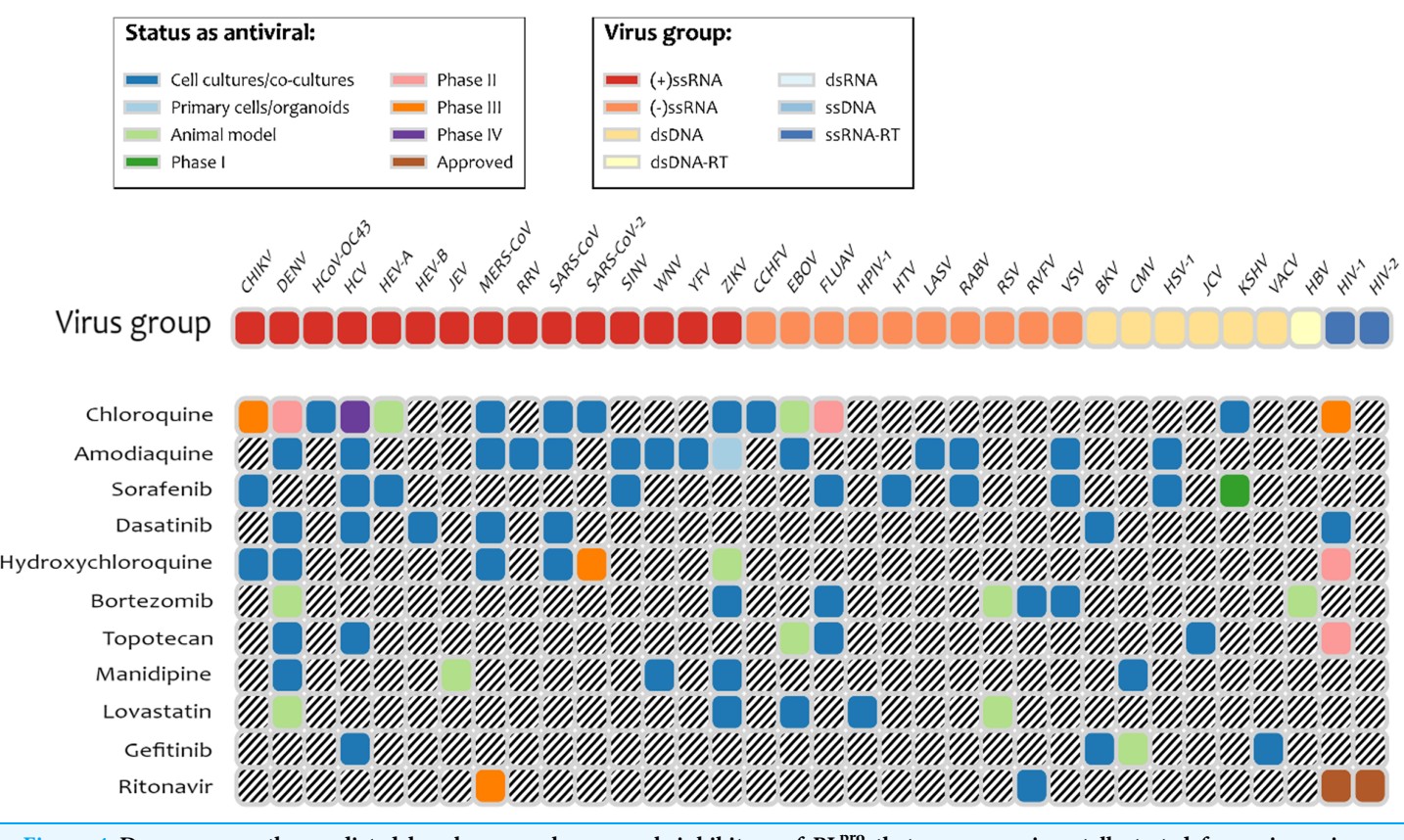

**Figure 4 Drugs among the predicted by pharmacophore search inhibitors of PL^pro that were experimentally tested for various viruses.** Obtained using DrugVirus.info database (*Zhang et al., 2020*).

binding energies on the same level with the random compound do not have to be completely discarded.

We are aware of two other studies where docking experiments were used to predict binding of existing pharmaceuticals to the SARS-CoV-2 PL^pro (*Devaraj et al., 2007*; *World Health Organization, 2020*). Both prior studies relied on homology modeling of part (*Arya et al., 2020*) or the entire SARS-CoV PL^pro. *Wu et al. (2020)* studied 2,924 compounds from ZINC Drug Database, as well as 78 known antivirals; while *Arya et al. (2020)* studied 2,525 FDA-approved compounds from DrugBank and the ZINC 15 database. Two compounds were identified in the present study and by *Wu et al. (2020)*: valganciclovir and pemextred. The remaining compounds identified here are unique to our study. This may reflect the influence of using the crystal structure of SARS-CoV-2 as the starting point in the present study, and a difference in methodology in our case including preliminary pharmacophore-based search before docking computational experiments.

It is interesting to note that several drugs with high docking energy were tested or are in experimental testing: nilotinib was active only for SARS-CoV (*Zhang et al., 2020*); dasatinib was confirmed to be active in cell-culture assays for MERS-CoV and SARS-CoV (*Osipiuk et al., 2020*). Dasatinib was also shown to be active against SARS-Cov-2 in clinical cases (*Abruzzese et al., 2020*). Terconazole and fluspirilene were shown to be active in

cell-culture assays for SARS-Cov-2 (*Abruzzese et al., 2020*). Manidipine was found in the database of experimental results for broad set of antiviral drugs, DrugVirus.info (*Andersen et al., 2020*). Indinavir and ritonavir (HIV viral protease inhibitor), boceprevir (Hepatitis C protease inhibitor), and valganciclovir (antiviral medication for CMV) were found with energies of binding to $PL^{pro}$ of −6.7 kcal/mol and better. We note that according to the DrugVirus.info database (*Andersen et al., 2020*), 11 of the compounds selected by the pharmacophore-based search showed activity against the set of viruses (Fig. 4) including amodiaquine, chloroquine, sorafenib, dasatenib, hydroxychloroquine, bortezomib, topotecan, manidipine, lovastatin, gefitinib, and ritonavir. Most experimental testing was done in cell-cultures, but there is also a significant amount of animal testing and several of these drugs are in different stages of clinical trials. The prior computational studies (*Harcourt et al., 2004*; *Arya et al., 2020*) did not identify any of these compounds as potential inhibitors of $PL^{pro}$, with the exception of chloroquine (*Arya et al., 2020*). On the other hand, *Wu et al. (2020)* identified two antivirals that our experiments did not predict as inhibitors: ribavirin and β-thymidine.

## ACKNOWLEDGEMENTS

We would like to thank the people of San Diego Supercomputer Center and CureMatch, Inc., for the friendly support. We also thank to the people of CCG (Montreal) and personally to Dr. Guillaume Fortin for the friendly support.

### Funding

The SDSC Comet supercomputer is supported by the NSF grant: ACI #1341698 Gateways to Discovery: Cyberinfrastructure for the Long Tail of Science. Mark Miller was supported by NIH R01 GM126463. Other authors received no funding for this work. The funders had no role in study design, data collection and analysis, decision to publish, or preparation of the manuscript.

### Grant Disclosures

The following grant information was disclosed by the authors:
NSF: ACI #1341698.
NIH: R01 GM126463.

### Competing Interests

Igor F. Tsigelny is a CSO of CureMatch Inc. The authors declare that they have no competing interests.

### Author Contributions

- Valentina L. Kouznetsova conceived and designed the experiments, performed the experiments, analyzed the data, prepared figures and/or tables, authored or reviewed drafts of the paper, developed a pharmacophore model, conducted clustering, and approved the final draft.

- Aidan Zhang conceived and designed the experiments, performed the experiments, analyzed the data, prepared figures and/or tables, authored or reviewed drafts of the paper, conducted computational docking, and approved the final draft.
- Mahidhar Tatineni conceived and designed the experiments, performed the experiments, analyzed the data, authored or reviewed drafts of the paper, conducted computational docking, and approved the final draft.
- Mark A. Miller conceived and designed the experiments, analyzed the data, prepared figures and/or tables, authored or reviewed drafts of the paper, and approved the final draft.
- Igor F. Tsigelny conceived and designed the experiments, analyzed the data, prepared figures and/or tables, authored or reviewed drafts of the paper, and approved the final draft.

## Data Availability

The raw data is available in Table S1.

## Supplemental Information

Supplemental information for this article can be found online at http://dx.doi.org/10.7717/peerj.9965#supplemental-information.

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
