# Peer review of "Potential COVID-19 papain-like protease PLpro inhibitors: repurposing FDA-approved drugs"

_PeerJ, doi:10.7717/peerj.9965_

## Round 0.1 · original submission · Major Revisions

Please provide a comprehensively revised version addressing the editorial comments and a detailed rebuttal letter.

Reviewer 1 ·

Basic reporting

no comment

Experimental design

no comment

Validity of the findings

no comment

Additional comments

The authors report that some commercial drugs are potential COVID-19 inhibitors using docking as a screening tool. The works is well addresses, nonetheless some comments should be kept in mind:

1) Line 63: add reference (see https://www.who.int/docs/default-source/coronaviruse/situation-reports/20200709-covid-19-sitrep-171.pdf?sfvrsn=9aba7ec7_2).

2) In the Methods section, the authors should provide the center and size (x,y,z) for the coordinates of the docking.

3) Current clinical therapy for COVID-19 is based on chloroquine, hidroxichloroquine, ivermectin and azithromycin. The authors should provide a discussion of the bonds formed and differences in the binding site of any of these compounds against the compounds found to be the best in docking to corroborate that they are better than the drugs used in clinical practice.

Reviewer 2 ·

Basic reporting

The manuscript discuss the discovery of new COVID-19 inhibitors. It is timely and important topic. However, Major concerns stands against positive response to the manuscript.

Experimental design

.General concerns
• The study focused on docking studies. However, Docking alone is not predictive to binding efficiency
• It is better to provide biological activity of the top 2-5 compounds
• MD simulation must be done including RMSD comparison, RMSF for residues, radius of gyration, mass movement and entropy dissection. MD is the most robust computational tool for detecting drug-protein complex stabilities.
• Rerank the compounds according to MMPSA /MMGBSA calculation and discuss the results in the light of it
• The used dataset is not well justified , especially the source of compounds and their molecular descriptors
• Provide the mutagenicity and carcinogenicity report for the compounds using the Korean Preadmet server

Validity of the findings

In Methods
I cannot understand how a pharmacophore model was generated from a structure does not contain any ligands. The used structure 6w9c is an Apo form of PLpro. Instead, the authors must reanalysis the pharmacophore model by using the new PLpro structures with modified amino acids ligands as 6WX4.
MOE was used in pharmacophore modeling, while autodock vina was used for docking. It is better to use MOE also in docking as it gives accurate results.
in results
The energy scores in Table 2 indicates few moderate binding compounds and the others are weak compounds
In Table 2, what are the clusters hh, jj, dd and S?
Discuss your finding in the light of previous PLpro virtual studies e.g. Pubmed PMID: 32597315, 32637945

Additional comments

Lines 40-42 are not accurate. The term protease comprises several families of proteins of diverse structure and function. The mentioned compounds must be accurately allocated to its corresponding protease e,g, Mpro, Plpro, …etc
Lines 44-45. These are host proteases. They also must be precisely described e.g. serine protease, …etc
Line 47: “Both proteases are highly conserved domains” it must be corrected , what is meant by domains?
Lines 51-55, mention the number and location of ORFs cut by PLpro.
Figures 1-8 can be merged into one file
Figures 9-10 could be moved to supplementary materials

Reviewer 3 ·

Basic reporting

Kouznetsova and co-authors present a nice study of a combination of ligand- and structure-based in silico approaches for the discovery of SARS-CoV-2 PLpro inhibitors.
The manuscript is a concise and very well written. The study showed the possible impact of massive computational power that a supercomputer can offer to researchers.
In my opinion, despite the absence of experimental validation of the identified hits, the paper give some nice insights into potential drugs that can be of use in compacting the current COVID-19 pandemic.
I have only a minor comments on the resolution of the protein structure used in the study (2.7A), the authors are advised to mention the resolution and comment on that such resolution although not optimal for computational efforts, it is of acceptable quality.
I recommend publishing the article in its current form.

Experimental design

No comments

Validity of the findings

No comments

---

## Round 0.2 · accepted · Accept

I concur with reviewer 1 and 3 in the assessment of the revised version. The suggestions have been taken into account and the manuscript is now accepted.